# Within-individual variation of C-reactive protein (CRP) measurements in primary care: A retrospective cohort study

Alex Gough[1]*, Alice Sitch[1,2], Tom Marshall[1]

1 Department of Applied Health Sciences, University of Birmingham, Birmingham, United Kingdom,
2 National Institute for Health and Care Research (NIHR) Birmingham Biomedical Research Centre, Birmingham, United Kingdom

* a.gough.2@bham.ac.uk

## Abstract

### Background

C-reactive protein (CRP) is a biomarker of inflammation used in diagnosis of inflammatory diseases and to guide treatment decisions. Variation in within-individual measured CRP may affect its clinical utility but estimates of within-individual variation are based on limited data and so may not be accurate.

### Methods

A retrospective cohort study was performed using data on CRP results and socio-demographic, lifestyle and comorbidity covariates extracted from the IQVIA Medical Research Database (IMRD) database using the DEXTER tool. A minimum of four measurements for each individual was the only inclusion criterion. CRP data were log-transformed for analysis. Within-individual measured variation was calculated as a coefficient of variation (CV) using a linear regression random effects model for the whole population and various subgroups.

### Results

472,811 participants were included in this study, making it the largest study of variation of CRP to date by a factor of approximately five. The overall coefficient of variation for CRP was 1.604 (95% CI 1.602 to 1.606). This is much higher than the median reported CV for CRP of previous studies which was 0.41. CV increased with patient median.

### Strengths and limitations

The large number of participants and the real-world nature of the results are important strengths of this study. Weaknesses included the problem of accounting for

**Data availability statement:** Data cannot be shared publicly because the authors do not have permission to share the data. IMRD data used for the study were obtained under licence from IQVIA; pseudonymised participant data are available from IQVIA subject to Scientific Review Committee approval. IMRD-UK data contains electronic health records from UK primary care. In compliance with the UK Data Protection Act and licensing agreements, the data cannot be shared via a public repository. These restrictions aim to protect patient confidentiality. The data underlying the results presented in the study are available from IQVIA (https://www.iqvia.com/locations/united-kingdom/solutions/life-sciences-industrysolutions/real-world-solutions/iqvia-medical-research-data).

**Funding:** The author(s) received no specific funding for this work.

**Competing interests:** The authors have declared that no competing interests exist.

confounding by indication, and the short half-life of CRP making it hard to distinguish between acute illness and physiological variation.

## Conclusions

Estimated within-individual variation in this analysis of real-world data is very high and is higher than previously reported. Variation increases with patient median CRP, that is with more severe disease status. This has important implications for the diagnosis, monitoring and clinical decision-making for inflammatory disease.

## Introduction

C-reactive protein (CRP) is an acute-phase protein synthesised in the liver, the plasma levels of which rise with inflammation [1]. It is a non-specific biomarker that aids with the diagnosis and assessment of severity of infectious and inflammatory diseases such as respiratory infections (including COVID-19) [2] and chronic inflammatory diseases such as rheumatoid arthritis [3,4]. It can help distinguish between aetiologies of disease, such as bacterial or viral pneumonia [5,6]. Low levels of CRP can be assessed with the high-sensitivity CRP (hsCRP) test which has been shown to be useful as marker for cardiovascular disease [7].

Current National Institute for Health and Care Excellence (NICE) guidelines state that if it is unclear after initial assessment whether a patient with a lower respiratory tract infection needs antibiotics, CRP testing can be used to help with decision-making. It is recommended that after a point of care CRP test, patients with CRP more than 100 mg/litre are given immediate antibiotics, patients with CRP less than 20 mg/L are not routinely provided with antibiotics, and consideration is given to providing patients with CRP 20–100 mg/L with backup antibiotics [8]. The American Gastroenterology Association guidelines for the role of biomarkers on the management of Crohn's disease suggest a cut-off of CRP of 5 mg/L, below which active inflammation is unlikely and in certain circumstances advise clinicians use this cut-off to decide whether endoscopic evaluation should be performed [9].

However, CRP levels, like all biological measurements, vary within an individual over time. This within-individual variation is broadly composed of three parts [10]. Pre-analytical variation is variation due to varying factors prior to a measurement, such as exercise, recent food or fluid intake or stress. Analytical variation is the variation introduced by imprecision in the measuring process. Biological variation is the within-individual variation, influenced partly by predictable factors such as season, time of day or monthly hormone cycles, but also to a large extent by chance. The sum of these three types of variation gives the total within-individual variation [10]. The higher the within-individual variation, the lower the probability that a single measurement is reflective of the true mean. In this study, it is not possible to separate analytical from biological variation, so within-individual coefficient of variations ($CV_T$) is taken to refer to total within-individual variation that is the sum of biological and analytical variation.

A database of biological variation exists that lists the variation of commonly measured clinical laboratory tests [11]. However, most of the data for this comes from trials with small numbers of participants, often measured in ideal conditions. A recent systematic review of the within-individual variation of CRP [12] found only 34 studies that reported this finding for CRP and 26 that reported it for hsCRP with a total of 92,031 subjects in all the studies combined (range 4–56,218). For CRP the median CV was 0.41 with a range of 0.11 to 0.89 and for hsCRP the median CV was 0.44 (range 0.27 to 0.76). However, many of these studies had small numbers of participants and/or small numbers of repeat measurements, with four out of five of the studies with more than 1000 patients only having two measurements of CRP. Little difference between subgroups based on health conditions was found in that paper but low numbers of studies, measurements and participants made it hard to draw conclusions.

## Aims

To use real-world data (that is data from clinical sources rather than experimental studies) from the IQVIA Medical Research Database (IMRD) to describe the within-individual variation of CRP in a large cohort of patients, and to describe any covariates that have an effect on the variation. To describe the probability of within-individual variation in CRP resulting in clinically important changes in CRP in order to inform clinical decision-making with regard to observed CRP changes in a patient.

## Methods

### Study design and data sources

A population-based retrospective cohort study was undertaken using data obtained from the IQVIA Medical Research Database (IMRD). IMRD incorporates data from The Health Information Network (THIN), a Cegedim Database. Reference made to THIN is intended to be descriptive of the data asset licensed by IQVIA. IMRD is a pseudanonymised database of the electronic healthcare records of patients registered with general practices in the UK which use compatible practice management systems. Data collection for the database commenced in 2003. As of March 2021, the database contained the records of over 20 million patients, of which around 4 million were active with their registered GP practice.

IMRD data has been shown to be generalisable to the general primary care population in terms of factors such as demographics, deprivation, condition prevalence and deaths [14]. The dataset used for this study consisted of 11,737,653 participants registered in 832 practices.

CRP tests in the database are the results of requests submitted from practices to central laboratories which have nationally mandated levels of quality control.

This study is reported according to the STrengthening the Reporting of Observational studies in Epidemiology (STROBE) statement for observational studies.

### Ethical approval

IQVIA Medical Research data has been approved by the NHS Health Research Authority (NHS Research Ethics Committee ref 18/LO/0441) for the purpose of medical and public health research and to supply data for researchers for scientifically approved studies. The use of the data for this study was approved by the IQVIA Scientific Review Committee on 19th November 2020 with the reference 20SRC068.

### Setting

Data was obtained from UK general practices registered with THIN/IMRD from database inception to 9th September 2021. Data extraction from the database was performed using the Dexter software [15] and was completed on 9th September 2021. The date of the latest entrance was 11th January 2021. The entire timeframe of the database from inception to last

entry was used for analyses except for sensitivity analysis by year, which included only the years 2010–2019, since years prior to 2010 and after 2019 were incomplete and contained only small numbers of measurements. The authors had no access to data that could identify individual participants.

## Participants

All participants in the database were eligible for inclusion. Except for sensitivity analyses, participants were excluded if they did not have at least four repeat measurements of CRP expressed in mg/L recorded. Four was chosen as the cut-off for repeat measurements for pragmatic reasons related to maximising the number of repeat measurements in the analysis while allowing a sufficiently large sample size for analyses. Participants were followed until the earliest date of either death, leaving the practice, stopping contributing to the database or the study end date.

Age given is age at time of first measurement.

## Variable choice

The primary outcome was within-individual variation of CRP calculated as the within-individual coefficient of variation ($CV_T$). For subgroup analysis, the dates of the CRP tests and additional variables including patient sociodemographic characteristics, lifestyle factors and diagnoses were extracted (see Table 1). Variables were selected as those hypothesised most likely to have an effect on variation based on clinical plausibility or that enabled calculation of variation (for example date of measurement). Variables for sensitivity analysis were chosen to assess factors which might affect the robustness of the results such as statistical methods, number of repeat measurements and unit of measurement.

**Table 1. Variables extracted from database.**

| Variables extracted | |
|---|---|
| Test characteristics | CRP result |
| | Date of measurement |
| | Unit of measurement |
| Patient characteristics | |
| Sociodemographic | Age |
| | Sex |
| | Ethnicity |
| | Townsend Deprivation Score |
| | BMI |
| | Geographical region |
| Lifestyle factors | Smoker status |
| | Alcohol consumption status |
| Diagnoses and comorbidities | Diabetes |
| | Hypertension |
| | Hyperthyroidism |
| | Hypothyroidism |
| | Ischaemic heart disease |
| | Heart failure |
| | Ischaemic stroke |
| | Haemorrhagic stroke |
| | All cancers |

## Data cleaning

CRP measurements are routinely expressed in mg/L, so other units of measurement were excluded. CRP must be 0 or above, therefore all negative values were excluded. From examination of a frequency distribution of the CRP values within the dataset, and published ranges of CRP values, values greater than 1000 were excluded as implausible [13]. Participants with missing CRP results were not eligible for inclusion. Where other comorbidities were not reported, they were assumed to be absent. Participants with missing data for other characteristics such as ethnicity, BMI or deprivation score were included in the main analysis but excluded from the subgroup analysis where these variables were used.

## Statistical methods

Since the CRP values were highly skewed, log-transformed data was used when calculating CVs [14]. To allow for log transformation of zero results, we used a very small correction value (adding 0.000001 to all results). Sensitivity analyses were performed without this correction which gave very similar results. The mixed command in Stata was then used to calculate within-individual variance using a random effects model. 95% confidence intervals for CVs were derived from the model in Stata. For consistency with other research, $CV_T$ was expressed as a decimal fraction. It was considered improbable that participants would have identical CRP results on four or more different occasions and that these were likely duplicated results, so these participants were excluded. If more than one measurement was performed on the same day, only the first result was included on the basis that others recorded for the same day were likely to be duplications or errors.

There is little published data on minimal clinically important difference (MCID), and this may be due to the highly skewed nature of CRP results, with vastly different results encountered in different disease processes and with differing disease severity. Results may be interpreted and perceived differently in different diseases. For example a 5 mg/L change in CRP level may be perceived very differently by a patient with rheumatoid arthritis and an initial CRP of 10 mg/L compared to a patient with an initial CRP of 150 mg/L with bacterial pneumonia. The probability of getting a second result of 10 mg/L (chosen arbitrarily as there is no consensus MCID) higher or lower than the initial result was calculated by counting the number of participants in which every combination of first and second results was achieved (for example 1st measurement result = 0–10, 2nd = 50–60 or 1st measurement result = 10–20, 2nd = 60–70 etc). The probability of each combination of first and second results was then calculated by dividing the number of participants with that combination by the total number of participants who had the first result.

Statistical code for Stata is available from the author on request.

# Results

## Participants

11,737,653 participants registered in 832 practices were available in the database for analysis. Fig 1 shows the flowchart of participant selection. 472,811 participants had at least four measures of CRP recorded as mg/L between 0 and 1000. The median number of measurements per individual was 10 (IQR 6–31).

## Descriptive data and outcome data

The overall coefficient of variation CRP was 1.604 (95% CI 1.602 to 1.606), which is consistent with a very high level of within-individual measured variation.

Table 2 shows the participant characteristics and main univariable subgroup analyses. 472,811 participants were included in the study, with a median age at first measurement of 68 (IQR 54–80), 34.6% being male. Missing ethnicity was recorded for 239,720/472,811 (50.7%) participants and Missing Townsend deprivation score for 66,658/472,811 (14.1%) participants. The median number of years between first and last measurement in the database was 6.7 (IQR 3.9 to 9.8).

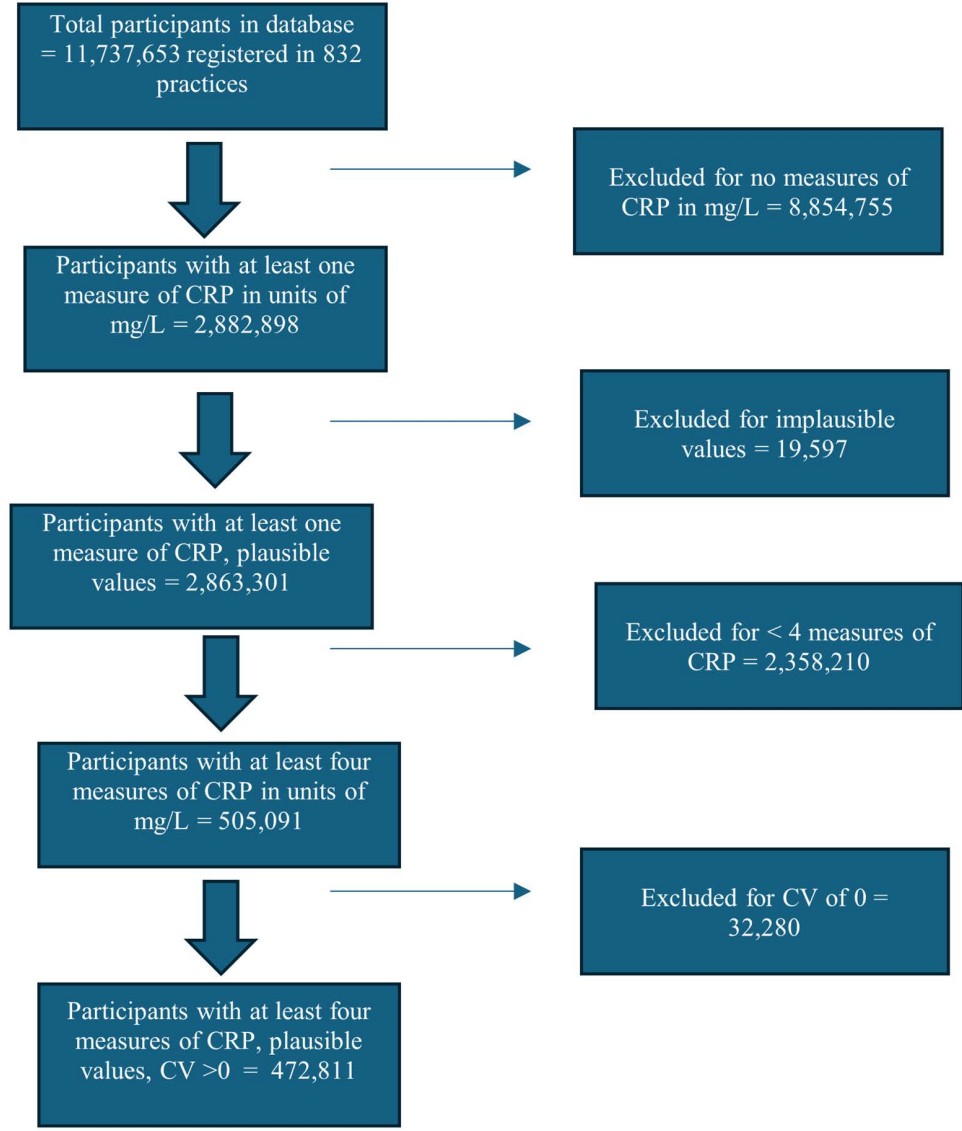

**Fig 1. Participant selection algorithm.**

**Subgroup analyses.** Subgroup analysis of $CV_T$ was also performed by sex, age, ethnicity, geographical region, Townsend deprivation score, BMI, presence of comorbidities, alcohol consumption and smoking status (see Table 2 and Appendix 1 Figs S1–S9 in S1 File). Higher $CV_T$s were seen in females than males, in white ethnicities compared to other (i.e., all non-white)ethnicities and in higher deprivation areas compared to lower deprivation areas.

Lower BMI was associated with a higher variation then higher BMIs. $CV_T$ tended to decrease with age, with a slight increase seen in the oldest participants. Compared to no comorbidity, patients with chronic obstructive pulmonary disease (COPD), chronic kidney disease (CKD) stage 3–5, heart failure, ischaemic heart disease, ischaemic stroke, ischaemic heart disease or cancer had a higher $CV_T$. Higher mean and median CRP levels were associated with higher variation.

**Table 2. Sociodemographic data and main univariable subgroup analyses. Other ethnicities includes all ethnicities not categorised as black, mixed, south Asian or white.**

| | | Median (mg/L) | CV(95% CI) | N(%) |
|---|---|---|---|---|
| Whole population | | 5.0 | 1.604 (1.602 to 1.606) | 472,811 (100) |
| Deprivation quintile (1 = least deprived 5 = most deprived) | 1 | 5.0 | 1.497 (1.494 to 1.501) | 97,832 (20.7) |
| | 2 | 5.0 | 1.408 (1.404 to 1.412) | 92,533 (19.6) |
| | 3 | 5.0 | 1.518 (1.513 to 1.522) | 90,681 (19.2) |
| | 4 | 5.0 | 1.379 (1.375 to 1.383) | 74,563 (15.8) |
| | 5 | 5.4 | 1.397 (1.392 to 1.403) | 50,524 (10.7) |
| | Missing | 5.0 | 2.975 (2.960 to 2.991) | 66,658 (14.1) |
| Ethnicity | Black | 5.0 | 1.576 (1.549 to 1.604) | 3,603 (0.8) |
| | Mixed | 4.0 | 1.244 (1.213 to 1.275) | 1,103 (0.2) |
| | Other | 4.0 | 1.613 (1.585 to 1.641) | 3,304 (0.7) |
| | South Asian | 4.0 | 1.721 (1.702 to 1.740) | 8,797 (1.9) |
| | White | 5.0 | 1.414 (1.412 to 1.417) | 216,282 (45.7) |
| | Missing | 5.0 | 1.778 (1.775 to 1.782) | 239,720 (50.7) |
| Sex | Male | 5.0 | 1.697 (1.693 to 1.701) | 163,718 (34.6) |
| | Female | 5.0 | 1.558 (1.556 to 1.561) | 309,093 (65.4) |
| Age (years) | <=16 | 2.0 | 2.575 (2.481 to 2.672) | 818 (0.2) |
| | 17–24 | 4.0 | 1.782 (1.754 to 1.809) | 5,729 (1.2) |
| | 25–34 | 4.0 | 1.592 (1.579 to 1.604) | 21,685 (4.6) |
| | 35–44 | 4.0 | 1.334 (1.326 to 1.342) | 34,790 (7.4) |
| | 45–54 | 4.5 | 1.247 (1.242 to 1.252) | 60,946 (12.9) |
| | 55–64 | 5.0 | 1.255 (1.251 to 1.259) | 81,718 (17.3) |
| | 65–74 | 5.0 | 1.182 (1.178 to 1.185) | 93,499 (19.8) |
| | >74 | 6.0 | 1.290 (1.287 to 1.293) | 173,626 (36.7) |
| Median CRP (mg/L) | 0 to <1 | 0.6 | 13.753 (13.438 to 14.077) | 14,668 (3.1) |
| | 1 to < 2 | 1.0 | 2.275 (2.265 to 2.286) | 61,504 (13.0) |
| | 2 to <3 | 2.0 | 1.737 (1.730 to 1.744) | 49,371 (10.4) |
| | 3 to < 4 | 3.0 | 1.521 (1.515 to 1.527) | 50,903 (10.8) |
| | 4 to <5 | 4.0 | 1.314 (1.310 to 1.319) | 53,354 (11.3) |
| | 5 to <6 | 5.0 | 1.175 (1.172 to 1.179) | 59,020 (12.5) |
| | 6 to < 7 | 6.0 | 1.424 (1.417 to 1.431) | 23,771 (5.0) |
| | 7 to < 8 | 7.0 | 1.321 (1.314 to 1.327) | 19,439 (4.1) |
| | 8 to < 9 | 8.0 | 1.224 (1.218 to 1.230) | 18,303 (3.9) |
| | 9 to < 10 | 9.0 | 1.165 (1.159 to 1.172) | 13,203 (2.8) |
| | 10 to <20 | 13.0 | 1.193 (1.190 to 1.196) | 65,110 (13.8) |
| | 20 to < 30 | 23.4 | 1.325 (1.318 to 1.331) | 20,051 (4.2) |
| | 30 to < 40 | 34.0 | 1.477 (1.464 to 1.602) | 8,949 (1.9) |
| | 40 to < 50 | 44.0 | 1.602 (1.583 to 1.622) | 5,051 (1.1) |
| | 50 to < 60 | 54.0 | 1.658 (1.631 to 1.687) | 2,996 (0.6) |
| | 60 to < 70 | 64.0 | 1.698 (1.662 to 1.735) | 1,915 (0.4) |
| | 70 to < 80 | 74.0 | 1.865 (1.813 to 1.918) | 1,351 (0.3) |
| | 80 to < 90 | 84.0 | 1.950 (1.885 to 2.019) | 952 (0.2) |
| | 90 to < 100 | 94.0 | 2.354 (2.245 to 2.472) | 667 (0.1) |
| | 100 to < 200 | 122.0 | 2.175 (2.117 to 2.235) | 2,038 (0.4) |
| | 200 to < 300 | 223.0 | 2.158 (1.966 to 2.381) | 160 (0.0) |
| | >=300 | 330.8 | 2.498 (1.776 to 3.827) | 16 (0.0) |

*(Continued)*

Median CRP level, deprivation score, ethnicity, sex and age had larger effects on CV, so these were chosen for multi-subgroup analysis. The results of this are presented in a heat map in Fig 2. It can be seen from this that of these subgroups, median CRP level seems to have the most marked effect on CV.

Fig 3 shows the CV by patient median CRP level, which demonstrates a marked increase in variation with patient increasing patient median.

Fig 4 shows a heat map with data on the probability of the result of a second CRP test given the result of an initial CRP test.

**Sensitivity analyses.** As the amount of missing data in ethnicity and Townsend deprivation score categories was high, a sensitivity analysis was carried out with participants without ethnicity or Townsend deprivation score dropped from the analysis. This analysis included 195,277 participants and gave a $CV_T$ 1.29 (95% CI 1.29 to 1.29) which is slightly lower than the whole population calculation.

$CV_T$ was calculated by year, quarter, number of measurements, patient mean CRP, median days between measurements, alternative methods of calculating CV (linear regression on non-log adjusted data, and a crude arithmetic method

| Sex | | Male | | | | | | Female | | | |
|---|---|---|---|---|---|---|---|---|---|---|---|
| Age | | 20-60 | | >60 | | 20-60 | | >60 | | | |
| Ethnicity | | White | Non-white | White | Non-white | White | Non-white | White | Non-white | | |
| CRP | Deprivation | | | | | | | | | | |
| 0 to 20 | 1 to 3 | 1.373 | 1.897 | 1.421 | 1.675 | 1.191 | 1.499 | 1.256 | 1.561 | | |
| | 4 to 5 | 1.344 | 1.434 | 1.305 | 1.615 | 1.161 | 1.398 | 1.209 | 1.572 | | |
| >20 to 100 | 1 to 3 | 1.685 | 1.935 | 1.602 | 1.696 | 1.282 | 1.180 | 1.404 | 1.466 | | |
| | 4 to 5 | 1.644 | 1.518 | 1.502 | 1.541 | 1.074 | 1.196 | 1.239 | 1.406 | | |
| >100 | 1 to 3 | 2.639 | 2.715 | 2.035 | 2.076 | 3.333 | 3.120 | 2.232 | 2.238 | | |
| | 4 to 5 | 2.360 | 2.429 | 1.866 | 2.507 | 2.033 | 2.445 | 1.777 | 1.960 | | |

**Fig 2. Heat map showing the coefficient of variation for multivariable analysis according to median CRP result, sex, age, ethnicity, deprivation and BMI.** Blue colours represent lower values of CV and red colours represent higher values of CV. Age is in years; CRP is in mg/L, deprivation score is Townsend deprivation score. Age < 20 excluded. Total N aged 20-60 = 142,970; Total N aged >60 = 261,566.

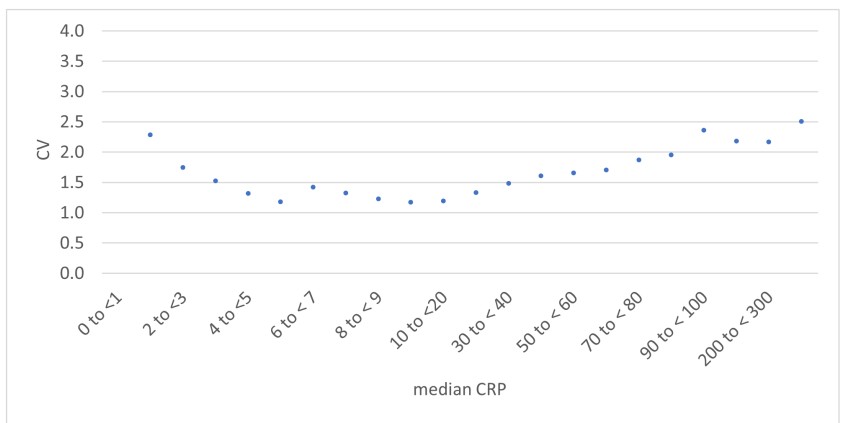

**Fig 3. Coefficient of variation by patient median CRP (mg/L) with 95% confidence intervals.**

| | second result | | | | | | | | | | | | | | | | | | | | Total probability of this difference | |
|---|---|---|---|---|---|---|---|---|---|---|---|---|---|---|---|---|---|---|---|---|---|---|
| first result | 0 to 10 | 10 to < 20 | 20 to < 30 | 30 to < 4( | 40 to < 50 | 50 to < 60 | 60 to < 70 | 70 to < 80 | 80 to < 90 | 90 to <100 | 100 to < 110 | 110 to <12( | 120 to <13( | 130 to <140 | 140 to <15( | 150 to < 160 | 160 to < 17( | 170 to < 180 | 180 to < 190 | 190 to < 200 | Difference 10 or less | Difference 10 or more |
| 0 to 10 | 81.70 | 8.47 | 2.82 | 1.57 | 1.04 | 0.76 | 0.57 | 0.48 | 0.35 | 0.30 | 0.26 | 0.21 | 0.18 | 0.14 | 0.13 | 0.11 | 0.11 | 0.09 | 0.07 | 0.06 | | 17.72 |
| 10 to < 20 | 47.83 | 32.53 | 8.10 | 3.43 | 1.87 | 1.20 | 0.94 | 0.55 | 0.54 | 0.38 | 0.35 | 0.31 | 0.21 | 0.18 | 0.15 | 0.14 | 0.11 | 0.11 | 0.09 | 0.10 | 47.83 | 18.75 |
| 20 to < 30 | 41.86 | 24.14 | 15.45 | 6.70 | 3.26 | 1.95 | 1.37 | 0.90 | 0.67 | 0.50 | 0.56 | 0.34 | 0.31 | 0.24 | 0.24 | 0.17 | 0.16 | 0.07 | 0.09 | 0.09 | 66.00 | 17.61 |
| 30 to < 40 | 41.45 | 18.64 | 13.17 | 9.59 | 5.09 | 3.13 | 2.08 | 1.27 | 1.02 | 0.85 | 0.69 | 0.43 | 0.28 | 0.30 | 0.30 | 0.22 | 0.19 | 0.13 | 0.10 | 0.09 | 73.26 | 16.18 |
| 40 to < 50 | 41.95 | 16.63 | 10.48 | 8.27 | 7.44 | 4.18 | 2.75 | 1.64 | 1.26 | 0.88 | 0.75 | 0.43 | 0.39 | 0.42 | 0.31 | 0.30 | 0.24 | 0.16 | 0.12 | 0.12 | 77.34 | 13.96 |
| 50 to < 60 | 41.36 | 16.33 | 9.22 | 7.65 | 5.89 | 5.55 | 3.95 | 2.59 | 1.66 | 1.12 | 0.77 | 0.57 | 0.49 | 0.47 | 0.32 | 0.37 | 0.20 | 0.20 | 0.15 | 0.05 | 80.44 | 12.90 |
| 60 to < 70 | 40.18 | 16.05 | 9.58 | 6.70 | 5.52 | 4.90 | 4.06 | 2.82 | 2.15 | 1.57 | 1.24 | 0.93 | 0.67 | 0.60 | 0.49 | 0.42 | 0.31 | 0.20 | 0.18 | 0.16 | 82.93 | 11.73 |
| 70 to < 80 | 40.06 | 15.34 | 8.79 | 5.71 | 5.06 | 4.80 | 4.12 | 3.95 | 2.82 | 1.84 | 1.64 | 0.88 | 0.79 | 0.79 | 0.59 | 0.37 | 0.51 | 0.17 | 0.23 | 0.31 | 83.87 | 10.93 |
| 80 to < 90 | 39.43 | 15.23 | 8.83 | 6.14 | 4.47 | 4.54 | 4.00 | 3.71 | 3.82 | 2.33 | 1.42 | 1.05 | 0.87 | 0.58 | 0.80 | 0.33 | 0.33 | 0.40 | 0.25 | 0.15 | 86.34 | 8.50 |
| 90 to <100 | 38.26 | 13.94 | 9.03 | 6.89 | 4.79 | 3.11 | 3.15 | 3.70 | 3.15 | 3.61 | 2.56 | 1.43 | 1.26 | 0.92 | 0.67 | 0.55 | 0.50 | 0.34 | 0.42 | 0.25 | 86.01 | 8.90 |
| 100 to < 110 | 35.20 | 16.63 | 8.68 | 6.44 | 4.88 | 4.05 | 3.22 | 3.51 | 2.88 | 2.93 | 2.78 | 1.56 | 1.12 | 1.51 | 0.83 | 0.39 | 0.49 | 0.88 | 0.24 | 0.39 | 88.40 | 7.41 |
| 110 to <120 | 33.98 | 15.45 | 8.05 | 5.68 | 4.68 | 5.45 | 4.14 | 2.66 | 2.96 | 2.43 | 2.96 | 3.43 | 1.95 | 1.30 | 1.01 | 0.83 | 0.77 | 0.30 | 0.59 | 0.24 | 88.45 | 6.99 |
| 120 to <130 | 32.82 | 16.09 | 9.42 | 5.34 | 5.27 | 4.57 | 3.30 | 2.60 | 2.81 | 2.18 | 2.39 | 2.18 | 1.90 | 1.83 | 0.70 | 1.48 | 0.91 | 0.98 | 0.42 | 0.49 | 88.97 | 6.82 |
| 130 to <140 | 35.15 | 13.56 | 8.03 | 4.84 | 5.35 | 3.80 | 2.94 | 3.11 | 2.68 | 2.33 | 2.85 | 2.25 | 2.25 | 3.02 | 1.73 | 0.95 | 1.04 | 0.86 | 0.78 | 0.60 | 89.12 | 5.96 |
| 140 to <150 | 33.13 | 13.05 | 8.56 | 4.89 | 4.18 | 4.38 | 4.79 | 3.77 | 2.55 | 1.73 | 2.55 | 1.53 | 1.12 | 2.14 | 2.45 | 1.73 | 1.94 | 0.51 | 0.71 | 0.51 | 88.38 | 5.40 |
| 150 to < 160 | 30.07 | 15.34 | 7.85 | 6.40 | 4.23 | 3.38 | 3.50 | 3.62 | 2.78 | 2.29 | 2.54 | 2.66 | 2.54 | 1.93 | 2.05 | 2.29 | 1.45 | 0.97 | 0.60 | 0.97 | 91.18 | 3.99 |
| 160 to < 170 | 30.98 | 14.86 | 8.19 | 6.68 | 4.41 | 3.40 | 2.52 | 3.02 | 2.52 | 2.77 | 1.89 | 1.64 | 1.39 | 2.52 | 1.39 | 1.26 | 2.14 | 1.51 | 1.76 | 0.63 | 89.42 | 3.90 |
| 170 to < 180 | 31.53 | 16.38 | 7.73 | 4.95 | 4.64 | 3.55 | 2.78 | 2.63 | 2.94 | 1.85 | 2.94 | 3.55 | 1.39 | 2.32 | 1.24 | 1.08 | 0.93 | 2.01 | 0.77 | 0.93 | 92.43 | 1.70 |
| 180 to < 190 | 28.91 | 13.90 | 9.58 | 6.07 | 4.79 | 4.31 | 3.83 | 2.40 | 2.24 | 2.40 | 1.60 | 1.92 | 1.28 | 2.08 | 1.28 | 1.76 | 1.44 | 1.44 | 2.72 | 1.12 | 91.21 | 1.12 |
| 190 to < 200 | 30.24 | 14.23 | 8.50 | 6.72 | 4.94 | 3.56 | 3.75 | 1.58 | 3.75 | 2.77 | 2.57 | 1.38 | 1.78 | 1.19 | 1.78 | 1.98 | 1.58 | 0.79 | 1.19 | 1.58 | 94.27 | |

**Fig 4. Probability of second result being 10 mg/L different than first result.** The last two columns show the probability (%) of the second result being at least 10 or more mg/L different than the first score. First and second results are in mg/L. All age groups were included in this analysis. Blue colours represent lower probabilities and red colours represent higher probabilities.


of calculating CV), and CV by patient mean on non-log adjusted data (see Appendix 2 Figs S10–S15 in S1 File). Little difference in CV was seen when stratified by year or quarter suggesting no trend over time in variation and no seasonal effect. An increased number of measurements was associated with a lower CV, perhaps because more measurements smoothed out spikes in CRP from transient illness. However, CV was markedly higher if the median time between measurements was less than about one month. There was little difference in CV when calculated by crude arithmetic methods and log-adjusted linear regression.

## Discussion

### Key results

This is the largest study of variation of CRP to date, with 472,811 participants, which is nearly five times more than the number of participants in all 60 published studies reporting CRP variation put together described in a recent systematic review [12].

The overall coefficient of variation for CRP was 1.604 (95% CI 1.602 to 1.606). This is much higher than the median reported for previous studies of 0.41 (range 0.11 to 0.89) [12]. The probability of a second measurement differing markedly from a first measurement is high as shown in Fig 4, although this finding must be interpreted in conjunction with the fact that CRP has a short half-life of around 19 hours after the inflammatory trigger is removed [15]. Furthermore, the data does not distinguish between situations where something has materially changed for example a change of medication or resolution of disease, and where nothing has changed.

Subgroup analyses showed that age, sex, ethnicity, BMI, and presence of certain comorbidities had an effect on $CV_T$ of CRP, with younger and older participants, female participants and participants with lower BMIs having higher $CV_T$s, likely due to a higher degree of pathological variation in these groups. Note that there may be some interaction between subgroups, with for example age, smoking, deprivation and BMI being associated with each other and with comorbidities. The median age was relatively high at 68 years which might be associated with a higher degree of pathological variation.

However, as the heat map of multiple subgroup analysis (Fig 2) and the graph of $CV_T$ versus median patient CRP (Fig 3) show, the effect of median patient CRP has a much larger effect than other subgroup analyses, with the $CV_T$ increasing markedly with increasing median patient CRP after a value of around 10 mg/L. This is important because patients with

more severe disease are likely to have a higher variation of results, making treatment and diagnosis decisions more difficult. For example at the cut-off point recommended by NICE for giving immediate antibiotics to patients with respiratory disease, 100 mg/L, the CV is roughly 50% higher than the overall population CV (2.35 versus 1.60).

## Strengths and limitations

The major strength of this study is the large size of the dataset which gives more accurate overall CV and allows the investigation of subgroups. It is also important that the data is real-world data taken from the clinical records of UK primary care practices, which are used for clinical decision-making, and not measurements taken under ideal or experimental conditions. The Biological Variation Data Critical Appraisal Checklist (BIVAC) recommends biological variation studies are performed on patients in steady state, with "preanalytical procedures described and standardized to minimize preanalytical variation" [16]. This is important when using biological variation data to set analytical performance specifications, but is less relevant to clinical decision-making. Thus the real-world nature of the data in this study can be considered a weakness from the point of view of setting analytical standards, but it is also a strength in that real-world data is the data used to make clinical decisions and it includes every component of variation (pre-analytical, analytical, biological and pathological).

Assumptions were made for the repeat results calculation, namely that the change between first and second result is representative of changes in the result in subsequent calculations, which may not be the case.

Another weakness with the data is that it is hard to account for confounding by indication, for example patients with greater disease severity are likely to have more frequent measurements. Nevertheless, the large disparity between this study and previously published results and the large number of participants help to compensate for this problem.

A further potential weakness with the data is error associated with data entry and recording into the clinical databases (although the electronic transmission of lab results directly into databases is likely to minimise this). Data cleaning prior to analysis also partially compensates for this. For example implausible values were removed (taking the cut-off points from analysis of the IMRD data itself and published data [13]), duplicate results on the same day were removed, participants with less than four repeated measurements were eliminated, and any patients with four or more identical results were removed, since this is a highly unlikely finding.

Inclusion of the entire dataset meant the only potential bias introduced into the results (that was not already present in the database) could come from data cleaning (for example exclusion of implausible values) and the specification that at least four measurements must be included. However, this latter specification, although aimed at increasing the accuracy of the variation calculation, could introduce confounding by indication since patients with more frequent tests might have more severe/less stable disease and hence a higher degree of pathological variation. Furthermore, it has been shown that the ordering of any test by a clinician increases the risk of the disease which the test is investigating, since the presence of a test in a patient record suggests there was a suspicion that patient was at risk of the disease. This could cause some degree of selection bias when compared to results obtained from healthy populations in controlled biological variation studies [17]. Analysing the mean duration of follow-up might also be helpful. Since age is taken as age at time of first measurement, there is some potential for misclassification of age given the long follow up for some participants.

The long follow-up means participants may have undergone disease or treatment transitions. Given CRP's short half-life, some of the variation may have reflected these transitions. The coefficient of variation in this study should therefore be considered as an upper-bound real-world estimate.

Finally the nature of CRP as an acute phase protein with a short-half life has implications for assessment of CRP variation over a long term. Fig S14 in S1 File shows that patients with a very short median interval between measurements have markedly higher variation, which is consistent with the change in CRP levels with acute disease progression and resolution and treatment. Nevertheless, the longer term within-individual variation (where the median interval between measurements is more than about a month) is quite consistent. Furthermore, CRP is used in the monitoring of chronic

disease, for example as a component of the DAS-28 instrument for monitoring rheumatoid arthritis [18] and so the long-term variation of CRP is clinically important.

## Interpretation and generalisability

This study shows that the variation of CRP in this population of primary care patients in the UK is much higher than results for variation previously reported in the literature [12]. This may be because the participants were less likely to be in a steady state, and the pre-analytical testing conditions were not rigorously controlled. However, as discussed, this is the data on which clinical decisions are based.

The findings of this study have clinical implications for diagnosis and monitoring. Not taking within-individual variation into account when making a diagnosis could lead to false positives and hence inappropriate treatment of for example rheumatoid arthritis with immunosuppressives leading to possible adverse effects, or bacterial pneumonia with antibiotics which could contribute to antimicrobial resistance. False negatives in diagnosis however could lead to delayed or missed opportunities for treatment, leading to poorer patient outcomes. Similarly, when using CRP for monitoring, for example of rheumatoid arthritis, a measured CRP value that is much higher than the real value may lead to inappropriate treatments with immunosuppressive drugs, increasing the risk of adverse effects, while a measured CRP value that is much lower than the real value may lead to treatment being withheld, causing patient morbidity and potential disease progression.

NICE guidelines recommend that CRP results are used to guide antibiotic treatment for acute lower respiratory tract disease. However, the high within-individual variation of CRP, especially at the higher values, could mean that a measured value is not reflective of the patient's true mean, so antibiotic treatment is inappropriately withheld. Similarly, AGA guidelines on the management of Crohn's disease recommend CRP is used to guide whether endoscopic investigation is required, and the high variation of CRP could mean that a patient is classified incorrectly as requiring or not requiring endoscopy [9].

The chart showing the probability of a second CRP result given the first result can be used to give some estimation of the extent of the problem. For example a patient with a first CRP measurement of 60–70 mg/L has an 82.9% chance of the second result being 10 mg/L lower than the first result and a 11.7% chance of the second result being 10 mg/L or more higher than the first result. The high chance of the second result being lower than the first may not only reflect variation, but some degree of regression to the mean, since diseases for which CRP are part of the diagnostic and monitoring process are often acute, and CRP has a short half-life. The fact that over 10% of patients show an important increase in CRP despite this fact helps confirm the high variation of CRP.

The CV calculated in this study can also be used to estimate the chance of getting a particular measured result given the true mean of a patient. For example, if we take a hypothetical patient with a true CRP of 60 mg/L (a value in the middle of the equivocal group for recommending an antibiotic for respiratory tract infection), the CV from the data in this study is approximately 1.6. This suggests a probability of 34% that the measured result will be 20 mg/L or lower, which would mean antibiotics were not prescribed, and a probability of 15% that the measured result will be 100 mg/L or higher, meaning that immediate antibiotics are recommended. Thus the high variation of CRP means that it is highly likely that single measurements will incorrectly categorise a patient when making treatment decisions.

Clinicians should be aware of the high within-individual variation of CRP and the high probability of incorrect inferences from CRP results that can result from this. They need to be aware of the possibility for misdiagnosis and inappropriate management of inflammatory and infectious conditions when relying on single measures of CRP, (although even taking multiple measurements does not completely eliminate this problem). Laboratories reporting results should alert clinicians to the high within-individual variability of CRP, perhaps using the CV to quantify the uncertainty around whether a result represents the patient's true mean. This is especially important when using CRP to make decisions about management of chronic diseases such as rheumatoid arthritis, where small differences in CRP might influence the decision whether

to use biologics. Authors of clinical guidelines should take variation into account when making diagnosis, monitoring and treatment recommendations. They should consider less definitive boundaries for classification of illness based on CRP, recommend repeat testing before making a diagnostic or management decision, and/or ensure clinicians are aware of the uncertainty that within-individual variation of CRP creates in a CRP measurement result.

Note that some of the underlying conditions for which CRP is part of the diagnostic or monitoring pathway are more variable than other conditions (for example bacterial pneumonia versus rheumatoid arthritis), and also more variable than other underlying conditions for which the tests examined in this thesis are performed. That is to say that the pathological variation in those conditions is higher in some conditions than others. Examples of conditions with a high degree of pathological variation include bacterial pneumonia and other acute illnesses, while those with a lower degree of pathological variation include chronic diseases such as rheumatoid arthritis and Crohn's disease.

Further research that addresses confounding by a change in frequency or timing of testing with increased disease severity or variation would be useful. It would be useful to perform a similar study restricted to patients with only long-term inflammatory diseases, to reduce the issue of confounding by fluctuation of the natural disease course. Studies into how variation in this population affects clinical outcomes such as mortality and whether reducing variation with improved testing and management protocols improves outcomes is also important. Other options for future research would be to work out the variation when patients are in a stable condition with respect to medication and disease.

## Conclusions

This study suggests that the total measured within-individual variation of CRP in real-world situations is much higher than previously reported in the literature. Within-individual variation of CRP increased with median CRP after values for CRP of around 10 mg/L.

## Supporting information

**S1 File. Tables S1-S3 and Figs S1-S15.**
(DOCX)

## Acknowledgments

Drs Krishna Ghokale, Dr Anuradhaa Subramanian, Dr Siang Ing Lee and Dr Naijie Guan for assistance with the use of Dexter to extract data from the IMRD and CPRD databases

## Author contributions

**Conceptualization:** Alex Gough, Alice Sitch, Tom Marshall.

**Data curation:** Alex Gough.

**Formal analysis:** Alex Gough.

**Investigation:** Alex Gough.

**Methodology:** Alex Gough, Alice Sitch, Tom Marshall.

**Project administration:** Alex Gough.

**Supervision:** Alice Sitch, Tom Marshall.

**Writing – original draft:** Alex Gough.

**Writing – review & editing:** Alex Gough, Alice Sitch, Tom Marshall.

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
