## [Decision Letter · Decision Letter 0]

22 Sep 2025

Dear Dr. Alex Gough,

Thank you for submitting your manuscript to PLOS ONE. After careful consideration, we feel that it has merit but does not fully meet PLOS ONE’s publication criteria as it currently stands. Therefore, we invite you to submit a revised version of the manuscript that addresses the points raised during the review process.

**ACADEMIC EDITOR: Major revision**

Please submit your revised manuscript by  Nov 06 2025 11:59PM. If you will need more time than this to complete your revisions, please reply to this message or contact the journal office at plosone@plos.org . A rebuttal letter that responds to each point raised by the academic editor and reviewer(s). You should upload this letter as a separate file labeled 'Response to Reviewers'.A marked-up copy of your manuscript that highlights changes made to the original version. You should upload this as a separate file labeled 'Revised Manuscript with Track Changes'.An unmarked version of your revised paper without tracked changes. You should upload this as a separate file labeled 'Manuscript'.

We look forward to receiving your revised manuscript.

Kind regards,

Marwan Salih Al-Nimer, MD, PhD

Academic Editor

PLOS ONE

Journal Requirements:

2. We noted in your submission details that a portion of your manuscript may have been presented or published elsewhere. [A version of this research made up a chapter in A. Gough's PhD thesis.] Please clarify whether this [conference proceeding or publication] was peer-reviewed and formally published. If this work was previously peer-reviewed and published, in the cover letter please provide the reason that this work does not constitute dual publication and should be included in the current manuscript.

Academic Editor

Lines 58-60: add references about the values of CRP

Lines 66-71: Add references about the type of variations

Line 117: Check each point of STROBE statement

Line 147: Mention the method that used in this study about missing data

Recheck the unit of CRP mg/L and mg/l

Line 181: there is no zero value for CRP, and a biotransformation way used in this study, i.e., log 0 is.......

Reviewers' comments:

Reviewer's Responses to Questions

**Comments to the Author**

1. Is the manuscript technically sound, and do the data support the conclusions?

Reviewer #1: Yes

Reviewer #2: Partly

2. Has the statistical analysis been performed appropriately and rigorously?

Reviewer #1: Yes

Reviewer #2: Yes

3. Have the authors made all data underlying the findings in their manuscript fully available?

Reviewer #1: Yes

Reviewer #2: Yes

4. Is the manuscript presented in an intelligible fashion and written in standard English?

Reviewer #1: No

Reviewer #2: Yes

Reviewer #1: General comments

• The study addresses an important aspect in the clinical interpretation of CRP levels for diagnosis and prognosis. The cohort is impressively large and well described. However, the very high missingness in ethnicity (50.7%) and deprivation (14.1%) limits subgroup interpretability (Table 2). Authors should clarify how this was handled and consider analyses restricted to complete cases, as the dataset remains large enough.

Methods and cohort description

• The flow diagram should be redesigned in a top-down, left-to-right waterfall format, with the number of participants excluded at each step and the justification for exclusion shown clearly on the horizontal branches.Exclusions currently listed in the Statistical Methods (lines 171–173) should be integrated into the flow chart to justify participant losses.

• The manuscript reports age at first CRP measurement, implying age stratification is based on baseline age. Given the 7-year median follow-up, participants likely transitioned into older categories and potentially contributed CRP levels in the incorrect age bracket. Authors should state clearly whether age was fixed at baseline or updated, and discuss potential misclassification.

• The long follow-up means participants plausibly underwent disease or treatment transitions. Given CRP’s short half-life (~19 hours), much variation likely reflects these transitions. Authors should clarify this in the Discussion and frame the coefficient of variation as an upper-bound real-world estimate, considering analyses that minimise transition effects.

• The manuscript mentions STROBE compliance but does not explain how the study size was determined. This should be included. Authors should also justify the inclusion criterion of ≥4 CRP measurements

Results

• Figures and tables should be fully captioned and units stated (e.g., Age in years, CRP in mg/L).

• In Table 2, variable-specific exclusions (age >100, CRP >300) create denominator inconsistencies. Authors should consider defining the analytic cohort a priori (age ≤100, CRP ≤300) so that all analyses share the same denominator, and apply exclusions consistently across all analyses with a clear rationale.

• Authors should also define what is meant by the category “Other” under ethnicity.

• Age grouping is inconsistent: Table 2 uses 10-year bands, some analyses use ≤10, while Figure 2 collapses to 20–60 vs >60, but do not necessarily align with physiological transitions or clinical relevance for interpreting CRP. CRP is strongly influenced by developmental stage, hormonal status, and comorbidity patterns. More meaningful grouping could strengthen the analysis and clinical interpretability. I suggest that authors harmonise age categories across all analyses. They could consider using commonly applied clinical or epidemiological bands.

• Figure 2 functions as a heatmap but lacks a colour key. Authors should add a legend or colour scale bar and state in the caption what the shading corresponds to.

• Supplementary Table S2 includes results for participants with only 2–3 CRP tests, although the inclusion criterion was ≥4. Authors should justify that.

• Discussion

• References to “Chapter 3” (lines 262, 265) should be removed and replaced with appropriate details in the manuscript.

• The Discussion section is comparatively brief, while the ‘Strengths and Limitations' section is relatively lengthy. This imbalance underemphasises interpretation, contextualisation, and clinical implications of the findings. Authors should expand the Discussion to integrate findings with prior studies, explore physiological and social demographic factors underlying the high within-individual CRP variability, and outline implications for guideline development and clinical decision-making.

The reference list requires revision to align with the PLOS ONE formatting guidelines. For example, reference 9 does

Reviewer #2: As I am not specialised in the field of blood tests and analyses, my comment concerns the need to include current references (if available).

I hope these suggestions will help you improve your work, which is already excellent.

I wish you every success.

**Do you want your identity to be public for this peer review?** For information about this choice, including consent withdrawal, please see our Privacy Policy

Reviewer #1: No

Reviewer #2: No

---

## [Author Response · Author response to Decision Letter 1]

31 Oct 2025

Dear Dr Chenette

Thank you for giving us the opportunity to revise our manuscript “Within-individual variation of C-reactive protein (CRP) measurements in primary care: a retrospective cohort study.” We would like to thank the reviewers for their thorough comments, which we think has made the paper much stronger. A point-by-point answer to their queries is appended below.

Yours sincerely,

Alex Gough MA VetMB PhD

Department of Applied Health Research

University of Birmingham

Author: The file names have been revised.

2. We noted in your submission details that a portion of your manuscript may have been presented or published elsewhere. [A version of this research made up a chapter in A. Gough's PhD thesis.] Please clarify whether this [conference proceeding or publication] was peer-reviewed and formally published. If this work was previously peer-reviewed and published, in the cover letter please provide the reason that this work does not constitute dual publication and should be included in the current manuscript.

Author: The work has not been peer-reviewed, published or presented as an abstract elsewhere except as a chapter in A. Gough’s PhD thesis.

Author: IMRD data used for the study were obtained under licence from IQVIA; pseudonymised participant data are available from IQVIA subject to Scientific Review Committee approval. IMRD-UK data contains electronic health records from UK primary care. In compliance with the UK Data Protection Act and licensing agreements, the data cannot be shared via a public repository. These restrictions aim to protect patient confidentiality. Requests can be submitted to bassam.bafadhal@iqvia.com.

Author: N/A

Academic Editor

5. Lines 58-60: add references about the values of CRP

Author: Added to manuscript

6. Lines 66-71: Add references about the type of variations

Author: Added to manuscript

7. Line 117: Check each point of STROBE statement

Author: Done

8. Line 147: Mention the method that used in this study about missing data

Author: Added to manuscript

9. Recheck the unit of CRP mg/L and mg/l

Author: Done

10. Line 181: there is no zero value for CRP, and a biotransformation way used in this study, i.e., log 0 is.......

Author: Thanks for pointing out this issue with log 0. On investigation, we discovered that results of 0 were treated as missing values by Stata after log transformation, leading to around 1% of participants being excluded. This led to a very small but noticeable difference in the results. As a consequence, the dataset has been reanalysed using the method described by Rock and Durbin (2003) which involves adding a small constant to the results prior to transformation.

Rocke DM, Durbin B. Approximate variance-stabilizing transformations for gene-expression microarray data. Bioinformatics. 2003 May 22;19(8):966-72. doi: 10.1093/bioinformatics/btg107. PMID: 12761059.

Reviewer #1: General comments

11. The study addresses an important aspect in the clinical interpretation of CRP levels for diagnosis and prognosis. The cohort is impressively large and well described. However, the very high missingness in ethnicity (50.7%) and deprivation (14.1%) limits subgroup interpretability (Table 2). Authors should clarify how this was handled and consider analyses restricted to complete cases, as the dataset remains large enough.

Author: A sensitivity analysis was performed with participants with missing ethnicity or deprivation – see sensitivity analysis in manuscript.

Methods and cohort description

12. The flow diagram should be redesigned in a top-down, left-to-right waterfall format, with the number of participants excluded at each step and the justification for exclusion shown clearly on the horizontal branches. Exclusions currently listed in the Statistical Methods (lines 171–173) should be integrated into the flow chart to justify participant losses.

Author: Fig 1 corrected.

13. The manuscript reports age at first CRP measurement, implying age stratification is based on baseline age. Given the 7-year median follow-up, participants likely transitioned into older categories and potentially contributed CRP levels in the incorrect age bracket. Authors should state clearly whether age was fixed at baseline or updated, and discuss potential misclassification.

Author: Corrected in manuscript.

14. The long follow-up means participants plausibly underwent disease or treatment transitions. Given CRP’s short half-life (~19 hours), much variation likely reflects these transitions. Authors should clarify this in the Discussion and frame the coefficient of variation as an upper-bound real-world estimate, considering analyses that minimise transition effects.

Author: Corrected in manuscript.

15. The manuscript mentions STROBE compliance but does not explain how the study size was determined. This should be included. Authors should also justify the inclusion criterion of ≥4 CRP measurements.

Author: Corrected in manuscript

Results

16. Figures and tables should be fully captioned and units stated (e.g., Age in years, CRP in mg/L).

Author: Corrected in manuscript

17. In Table 2, variable-specific exclusions (age >100, CRP >300) create denominator inconsistencies. Authors should consider defining the analytic cohort a priori (age ≤100, CRP ≤300) so that all analyses share the same denominator, and apply exclusions consistently across all analyses with a clear rationale.

Author: Corrected in manuscript

18. Authors should also define what is meant by the category “Other” under ethnicity.

Author: Addressed in manuscript

19. Age grouping is inconsistent: Table 2 uses 10-year bands, some analyses use ≤10, while Figure 2 collapses to 20–60 vs >60, but do not necessarily align with physiological transitions or clinical relevance for interpreting CRP. CRP is strongly influenced by developmental stage, hormonal status, and comorbidity patterns. More meaningful grouping could strengthen the analysis and clinical interpretability. I suggest that authors harmonise age categories across all analyses. They could consider using commonly applied clinical or epidemiological bands.

Author: Age grouping altered for table 2 and fig 2.

20. Figure 2 functions as a heatmap but lacks a colour key. Authors should add a legend or colour scale bar and state in the caption what the shading corresponds to.

Author: Colour key added to legend.

21 Supplementary Table S2 includes results for participants with only 2–3 CRP tests, although the inclusion criterion was ≥4. Authors should justify that.

Author: Justification added to legend for Fig S12

Discussion

22. References to “Chapter 3” (lines 262, 265) should be removed and replaced with appropriate details in the manuscript.

Author: Corrected in manuscript

23. The Discussion section is comparatively brief, while the ‘Strengths and Limitations' section is relatively lengthy. This imbalance underemphasises interpretation, contextualisation, and clinical implications of the findings. Authors should expand the Discussion to integrate findings with prior studies, explore physiological and social demographic factors underlying the high within-individual CRP variability, and outline implications for guideline development and clinical decision-making.

Author: Discussion revised in manuscript

24. The reference list requires revision to align with the PLOS ONE formatting guidelines. For example, reference 9 does

Author Reference list corrected.

Reviewer #2:

As I am not specialised in the field of blood tests and analyses, my comment concerns the need to include current references (if available).

I hope these suggestions will help you improve your work, which is already excellent.

I wish you every success.

Author: A new reference [4] was added.

---

## [Editor Report · Decision Letter 1]

5 Nov 2025

Within-individual variation of C-reactive protein (CRP) measurements in primary care: a retrospective cohort study

PONE-D-25-45864R1

Dear Dr. % Alex Gough,

We’re pleased to inform you that your manuscript has been judged scientifically suitable for publication and will be formally accepted for publication once it meets all outstanding technical requirements.

Kind regards,

Marwan Salih Al-Nimer, MD, PhD

Academic Editor

PLOS ONE
---

## [Editor Report · Acceptance letter]

PONE-D-25-45864R1

PLOS ONE

Dear Dr. Gough,

I'm pleased to inform you that your manuscript has been deemed suitable for publication in PLOS ONE. Congratulations! Your manuscript is now being handed over to our production team.

Kind regards,

on behalf of

Professor Marwan Salih Al-Nimer

Academic Editor

PLOS ONE